# Opioid-Sparing Effect of Multi-Point Incision-Based Rectus Sheath Block in Laparoscopic-Assisted Radical Gastrectomy: A Randomized Clinical Trial

**DOI:** 10.3390/jcm12041414

**Published:** 2023-02-10

**Authors:** De-Wen Shi, Xiao-Dan Zhou, Feng-Jie Wang, Jing Wang, Yang Liu, Yong Niu, Guang-Hong Xu

**Affiliations:** 1Department of Anesthesiology, The First Affiliated Hospital of Anhui Medical University, 218 Jixi Road, Hefei 230022, China; 2Key Laboratory of Anesthesia and Perioperative Medicine of Anhui Higher Education Institutes, 218 Jixi Road, Hefei 230022, China

**Keywords:** laparoscopic-assisted radical gastrectomy, rectus sheath block, multi-point injection, ropivacaine, remifentanil, postoperative pain

## Abstract

Background: Profound trauma from laparoscopic-assisted gastrectomy (LAG) requires medication with a large number of opioids. The purpose of our study was to observe whether an incision-based rectus sheath block (IBRSB) based on the locations of the surgical incision could significantly reduce the consumption of remifentanil during LAG. Methods: A total of 76 patients were included. The patients were prospectively randomized into two groups. Patients in group IBRSB (*n* = 38) received ultrasound-guided IBRSB, and the patients received 0.4% ropivacaine 40–50 mL. Patients in group C (*n* = 38) received the same IBRSB with 40–50 mL normal saline. The following were recorded: the consumption of remifentanil and sufentanil during surgery, pain scores at rest and during conscious activity in the post-anesthesia care unit (PACU) and at 6, 12, 24, and 48 h after surgery, and use of the patient-controlled analgesia (PCA) at 24 and 48 h after surgery. Results: A total of 60 participants completed the trial. The consumption of remifentanil and sufentanil in group IBRSB were significantly lower than that in group C (*p* < 0.001). Pain scores at rest and during conscious activity in the PACU and at 6, 12, 24, and 48 h after surgery and patients’ PCA consumption within 48 h of surgery were significantly lower in group IBRSB than in group C (all *p* < 0.05). Conclusions: IBRSB based on incision multimodal anesthesia can effectively reduce the consumption of opioids during LAG, improving the postoperative analgesic effect and increasing patients’ satisfaction.

## 1. Introduction

Laparoscopic-assisted radical gastrectomy (LAG) is an effective surgery for the treatment of gastric cancer [1]. Although LAG is less traumatic than open surgery for the patient, postoperative incision pain is still prominent. In addition, LAG surgery still requires a large number of opioids to maintain anesthesia. Remifentanil, an ultra-short-acting opioid, is widely used in intraoperative analgesia [2]. However, intraoperative high-dose remifentanil might be related to postoperative hyperalgesia and increased postoperative opioid consumption [3].

In recent years, with the popularization of ultrasound, the ultrasound-guided nerve block has become an important part of multimodal analgesia. Studies have shown that the transverse abdominal plane block (TAPB) or rectus sheath block (RSB) can significantly reduce the use of intraoperative and postoperative analgesics and reduce postoperative pain scores in patients with abdominal median incision surgery [4,5,6,7,8]. However, some controversial research has found that there were no differences, and that the effects were not obvious compared to the control group [9,10]. The reasons for these inconsistent results may be the inconsistent dosage or concentration of the drugs used. In addition, the multiple and dispersed incisions in laparoscopic surgery may also be a contributing factor. Therefore, using only bilateral TAPB or RSB may make it difficult to effectively block incision pain [9,10]. However, an incision-based rectus sheath block (IBRSB) may be able to achieve the dual effect of nerve block and local invasiveness, because the block sites are closer to the surgical incision than TAPB or RSB alone. We therefore designed a prospective, randomized, double-blind study to observe whether IBRSB could reduce intraoperative opioid use, promote patients’ rapid postoperative return to consciousness, and improve postoperative analgesia.

## 2. Materials and Methods

This trial was registered in the Chinese Clinical Pathway Registry (ChiCTR2000038846) prior to study initiation. The ethics committee of our hospital (PJ202012-20) provided ethical approval for this study. The patients in this study provided written consent.

In total, 84 patients with American Society of Anesthesiologists grades Ⅱ–Ⅲ, aged 18–75 y, and who were about to undergo LAG were screened. Exclusion criteria were contraindications to RSB, such as coagulopathy, infection at the puncture site, preoperative cognitive impairment, a mental or language barrier; regular opioid usage; the surgical incision or trocar insertion being clearly beyond the range of the rectus sheath; chemotherapy prior to surgery; a change of surgeon; refusal to participate; and any condition that the investigator determined would adversely affect the study. A total of 76 patients participated in the study, as shown in Figure 1. A computer-generated allocation program randomly assigned the patients to group IBRSB (IBRSB with 0.4% ropivacaine) and group C (IBRSB with normal saline), and the final 60 patients completed the study; the consolidated standards of reporting trials (CONSORT) diagram is included here (Figure 1). The results are hidden in opaque envelopes that can only be opened with consent. None of the patients, surgeons, anesthesiologists, or follow-up researchers were aware of the grouping.

### 2.1. Study Protocol

All of the patients received standard monitoring, including invasive blood pressure (IBP), pulse oximetry (SPO2), and electrocardiogram (ECG) recordings. General anesthesia was induced with midazolam (0.02–0.05 mg/kg), etomidate (0.2–0.3 mg/kg), sufentanil (0.3–0.5 μg/kg), and cis-atracurium (0.2–0.3 mg/kg), followed by mask ventilation for 3 min and intubation. A patient end-tidal carbon dioxide of 35–45 mmHg was maintained. Propofol and remifentanil were used to maintain anesthesia. The bispectral index (BIS) was maintained at 40–60, and either blood pressure was maintained at ±20% of the basal value, or systolic blood pressure (SBP) was maintained at <160 mmHg. During the surgery, a heart rate (HR) or SBP > 20% of the baseline value for 1 min and/or BIS > 60 were considered as insufficient anesthesia [11,12]. When BIS > 60, propofol was given intravenously bolus at a dose of 0.5 mg/kg, and then the infusion rate of propofol was adjusted until the BIS range was between 40–60 [11]. For hypertension or tachycardia, remifentanil was given as an intravenous bolus at a dose of 0.5 μg/kg, followed by an increasing in the rate of remifentanil infusion. The infusion rate of remifentanil was increased in the range of 0.5 μg/kg/h to 1 μg/kg/h and adjusted once every 15–20 s until anesthesia was satisfactory. When the infusion dose of remifentanil exceeded 1 mg/h, 5–10 μg of sufentanil and/or vasoactive drugs were injected [11]. On the contrary, when blood pressure stabilized at ±20% of the basal value, the infusion dose of remifentanil was gradually reduced according to changes in blood pressure until the infusion was stopped. When hypotension (SBP < 20% of the baseline value for 1 min) was present, an intravenous bolus of 3–6 mg of ephedrine was given, and the infusion rate of the lactated ringer solution was increased; for bradycardia (HR < 45 bpm), an intravenous bolus of 0.2–0.5 mg of atropine was given. In addition, all of the patients received 0.7 minimum alveolar concentration (MAC) sevoflurane, which was administered to prevent intraoperative awareness, and all of the patients received 50 ug of dexmedetomidine at a rate of 0.1–0.3 μg/kg/h at the beginning of anesthesia until 30 min before the end of surgery. Prior to incision making and 30 min before the end of surgery, 50 mg of flurbiprofen axetil and 5–10 µg of sufentanil were given, and patient-controlled analgesia (PCA: the drug consisted of 2.5 ug/kg of sufentanil, 100 mg of flurbiprofen axetil, and NS, with a total dose of 100 mL) was then connected to the patients. Routine analgesia was administered using 100 mg of flurbiprofen axetil ester daily for 3d postoperatively. PCA was opened with 3 mL bolus, and infusion at 2 mL/L was continued when the first postoperative visual analogue scale (VAS) score exceeded 3, or if the patient made a request. A rescue analgesic drug of 50–100 mg of flurbiprofen axetil was given if pain persisted or if the patients required it.

After general anesthesia induction, all of the patients received IBRSB; IBRSB was divided into four injections, and the locations of the surgical incision were determined by the surgeon. IBRSB were performed according to a surgical incision under ultrasound (M-Turbo; FUJIFILM Sonosite Inc., Bothell, WA, USA), as shown in Figure 2. When the puncture needle reached the posterior sheath of the rectus abdominis muscle, 2 mL of normal saline was injected first after no air and no blood was withdrawn. The drug was injected after the water separated and the location of the tip was confirmed; the remaining three injection sites performed the same procedure. The patients in group IBRSB received 40–50 mL of 0.4% ropivacaine, and patients in group C, 40–50 mL of normal saline. Obvious drug diffusion was observed under ultrasound, indicating a successful block, as shown in Figure 3. All of the IBRSB and surgical procedures were performed by the same team of anesthesiologists and surgeons. There was a 15 min interval between IBRSB and skin incision.

### 2.2. Outcome Measures

Our primary endpoint was remifentanil consumption during surgery. Our secondary endpoints were VAS scores at rest and during conscious activity in the PACU and at 6, 12, 24, and 48 h after surgery, and the dosage of patient-controlled analgesia (PCA) used at 24 and 48 h after surgery. We also recorded the time of eye opening, extubation, PACU stay duration, first ambulation, and the first flatus in each group after surgery, and the consumption of postoperative rescue analgesic drugs and of sufentanil, and propofol, intraoperative fluid infusion, the duration of surgery and anesthesia, side effects, postoperative hospital stay and patient satisfaction, using a Likert 5-level scale (1 being very dissatisfied and 5 being very satisfied).

### 2.3. Statistical Analysis

SPSS version 16 software (IBM Corp., Armonk, NY, USA) was used for all of the statistical data analysis. The significance level was set at *p*-value < 0.05. The sample size was calculated from a pilot study of 20 patients randomized to group IBRSB or C. The respective mean values were 851.7 and 1356.5 μg. The respective standard deviations were 564.5 and 577.7 μg. At least 58 cases, 29 patients per group, could provide 90% power with a two-sided α of 5% to detect a significant difference between the groups. Considering the loss rate, the 76 patients was considered to be adequate.

The mean and variance were used to represent continuous variables, and the independent sample T-test was used to analyze the data consistent with normal distribution; otherwise, the median and range were used to represent the data, and the Mann–Whitneyu U test was used to analyze the data. The categorical variables were represented by percentages or numbers and analyzed using Pearson Chi-square test or Fisher’s exact test. 

## 3. Results

From January to May 2021, we evaluated the study eligibility for 84 patients scheduled for LAG treatment, and eight were excluded; three did not meet the inclusion criteria (the surgeons were changed in three), and five declined to participate. The remaining 76 patients were randomly assigned to group IBRSB and C. In group IBRSB, seven patients withdrew from the study during the study (there were two patients who changed surgical methods, three on whom extensive lymph node dissection was performed, and two found to have severe hypotension), and one patient was lost to follow-up during the postoperative follow-up period. In group C, six patients withdrew from the study during the study (there were four patients who changed surgical methods, one on whom extensive lymph node dissection was performed, and one found to have severe hypotension) and two patients declined to continue to participate in the study. A final total of 60 patients completed the study (Figure 1). Baseline characteristics showed no difference between the groups (Table 1). There was no statistical difference in the proportion of the three surgical methods (laparoscopic-assisted radical distal gastrectomy, laparoscopic-assisted proximal gastrectomy, laparoscopic-assisted total gastrectomy) between the groups (Table 1). There was no statistical difference in the type of gastric cancer between the two groups (Table 1).

As shown in Table 2, the consumption of remifentanil and sufentanil in group IBRSB was significantly lower than in group C (*p* < 0.001 and *p* < 0.001, respectively). There were no significant differences in propofol consumption (*p* = 0.211), the time of surgery and anesthesia (*p* > 0.05), and the intraoperative crystal infusion volume between both groups (*p* > 0.05); however, the amount of colloid infusion in group C was more than that in group IBRSB (*p* = 0.042). As shown in Table 3, the time of eye opening, extubation, and PACU stay in group IBRSB were shorter than that in group C (*p* = 0.007, *p* = 0.033, and *p* = 0.031, respectively). VAS scores at rest and during conscious activity in the PACU and at 6, 12, 24, and 48 h after surgery were significantly lower in the patients in group IBRSB (all *p* < 0.05). The PCA sufentanil consumptions at 24 and 48 h after surgery were also significantly lower in group IBRSB (*p* < 0.001). There was no significant difference in flurbiprofen axetil consumption between the two groups on the day of the surgery and on the first and second days after surgery (all *p* > 0.05). The time of the first flatus after surgery was shorter in group IBRSB (*p* < 0.05). There was no significant difference in postoperative hospital stay and the time of the first ambulation (*p* > 0.05). More patients in group IBRSB than in group C were satisfied with the anesthesia protocol (*p* < 0.001).

## 4. Discussion

In this study, results show that patients receiving the IBRSB had a lower intraoperative consumption of remifentanil; the IBRSB promoted rapid postoperative recovery of patients, improving the postoperative analgesic effect and increasing patient satisfaction. In addition, the VAS score and the PCA consumption of patients within 48 h of surgery in group IBRSB were significantly lower than those of group C; the time of the first flatus after surgery in group IBRSB was shorter than in group C. There was no significant difference in the first time of ambulation and postoperative hospital stay between both groups.

Gastric cancer is currently the fifth most common cancer, with more than 1 million cases in 2018 [13]. LAG has been increasingly used in patients with gastric cancer. Our study observed whether IBRSB could spare opioid use in LAG. Although laparoscopic surgery is significantly less traumatic than open surgery, a large amount of opioids are still needed to maintain anesthesia, and postoperative incision pain is still prominent. In recent years, after the popularization of ultrasound in clinical practice, TAPB, RSB, and other nerve blocks have been widely used in abdominal surgery, and can improve postoperative pain [6,14,15]. 

During LAG surgery, a midline incision is required, and there are several trocar insertions on both sides of the incision. RSB is a simple anterior abdominal wall block technique which blocks the anterior cutaneous branches of spinal nerves T7–T12 by distributing local anesthetics between the rectus abdominis muscle and its posterior sheath. RSB has been widely used in the repair of umbilical hernias in pediatric patients since the 1900s [6], and its advantages in reducing the consumption of intraoperative opioids and relieving postoperative pain have been widely recognized [6]. Since pediatric umbilical hernia repair only involves incision pain, RSB can be advantageous; other research also shows that RSB plays an important role in anterior abdominal wall surgery involving visceral pain [4,5,16]. Studies have reported that in laparoscopic surgery, bilateral RSB can reduce postoperative pain and opioid consumption in the early postoperative period [4,8]. Maloney et al. found that RSB leads to decreased opioid consumption and that patients who had RSB had lower pain scores compared to local infiltration analgesia in pediatric single incision laparoscopic surgery [17]. However, Kinjo et al. found no difference in the reduction of postoperative pain in RSB compared to the control group in gynecologic laparoscopy [9]. TAPB is also widely used for analgesia during and after abdominal surgery [14,18,19]. Oh et al. have also reported that TAPB did not have a significant effect on postoperative pain control in the early (0–2 h) and late (24 h) periods at rest after laparoscopic surgery [10]. Possible reasons for these may be related to the dose and concentration of drugs, the large trauma due to surgery, and the many and scattered incisions of laparoscopic surgery. 

Bloc et al. reported that an incision-based nerve block could reduce the consumption of remifentanil during cardiac surgery [11]. Cheng et al. reported that an incision-based nerve block can reduce the postoperative VAS score and improve postoperative analgesia in thoracic surgery [20]. According to these studies, based on the intraoperative incision and the location of the trocar insertions, we performed a bilateral multi-point block to ensure good diffusion of local anesthetic throughout the posterior rectus sheath to meet the needs of multi-point and dispersed surgical incisions. Regional block and local infiltration analgesia are important components of multimodal analgesia. We found that this modified IBRSB could achieve dual effects on nerve block and local infiltration anesthesia of the trocar insertions and the middle incision. Additionally, the posterior rectus sheath is closer to the peritoneum than the TAPB, and local anesthetics can partially penetrate the peritoneum by injecting local anesthetics into the posterior rectus sheath. We hypothesize that the IBRSB could produce a partial block effect on the supraperitoneal nerve endings, while the TAPB may find it more difficult to produce the block effect. 

Our results showed that the IBRSB could significantly reduce the consumption of remifentanil during surgery. Our results confirmed those of related studies [11,21]. Remifentanil, as a μ-opioid agonist, is often used as an intraoperative high-dose continuous infusion in LAG because of its rapid onset and predictable metabolism [2]. The intraoperative use of remifentanil has been shown to cause acute opioid tolerance and hyperalgesia, as well as increased postoperative pain and analgesic requirements [3]. For postoperative pain management of patients, the opioid tolerance caused by remifentanil can be overcome by increasing the dose of analgesics. However, in turn, this promotes the hyperalgesia effects of opioids. Therefore, we hope to reduce the consumption of remifentanil during operations by adding IBRSB to LAG anesthesia management. Our results also showed that this anesthesia regimen was effective.

Our results revealed that the nerve block could shorten the PACU stay time and extubation time. This may be because the nerve block group reduced intraoperative opioid use and promoted rapid recovery after surgery.

Similarly, PCA consumption and pain scores within 48 h of surgery in group IBRSB were significantly lower than those in group C. Our results showed that IBRSB administered to incisions improved postoperative analgesia. The results of previous studies were consistent with our current results [4,8]. The results of our study showed that there was no significant difference in the amount of postoperative rescue analgesic drug needed, which was different from the results of a study by Hamid et al., possibly because our sample size was small [8]. 

In terms of postoperative recovery, the results of this study showed that the nerve block group could shorten the time of the first flatus after surgery, which may be due to the nerve block group reduction of the use of opioids and promotion of recovery of postoperative gastrointestinal function. There were no significant differences in the time of the first ambulation, postoperative hospital stay, and incidence of postoperative adverse reactions between the two groups, but this may also be due to the small sample size of our study.

There were still several limitations that should be mentioned. First, the sample size was estimated according to the results about the remifentanil consumption of the preliminary test, and the small sample size may be the reason that there were no significant differences in postoperative rehabilitation, nausea and vomiting between the two groups. Second, in our study, IBRSB group significantly reduced intraoperative opioid use and improved postoperative analgesia compared with control, but this study did not investigate whether IBRSB was superior to bilateral RSB. We plan to further investigate the difference between the IBRSB and the bilateral RSB in the future. We plan to further investigate the difference between the approach we used and the traditional approach in the future. Thirdly, in our study, the consumption of remifentanil was the primary endpoint, and we adjusted its infusion rate according to intraoperative blood pressure fluctuations. In future studies, we plan to also use nociception index monitoring to adjust the intraoperative remifentanil infusion rate to observe the efficiency of our multi-point rectus abdominis sheath block. Therefore, caution should be taken in the generalization of conclusions.

## 5. Conclusions

Our research revealed that the IBRSB multi-modal anesthesia strategy could effectively reduce the consumption of remifentanil in patients undergoing laparoscopic-assisted radical gastrectomy, promote rapid recovery after surgery, and improve postoperative analgesia. This multi-modal anesthesia strategy has potential benefits for such patients.

## Figures and Tables

**Figure 1 jcm-12-01414-f001:**
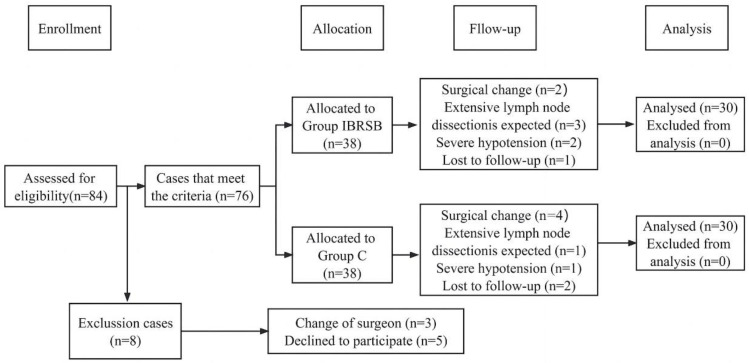
The CONSORT flowchart.

**Figure 2 jcm-12-01414-f002:**
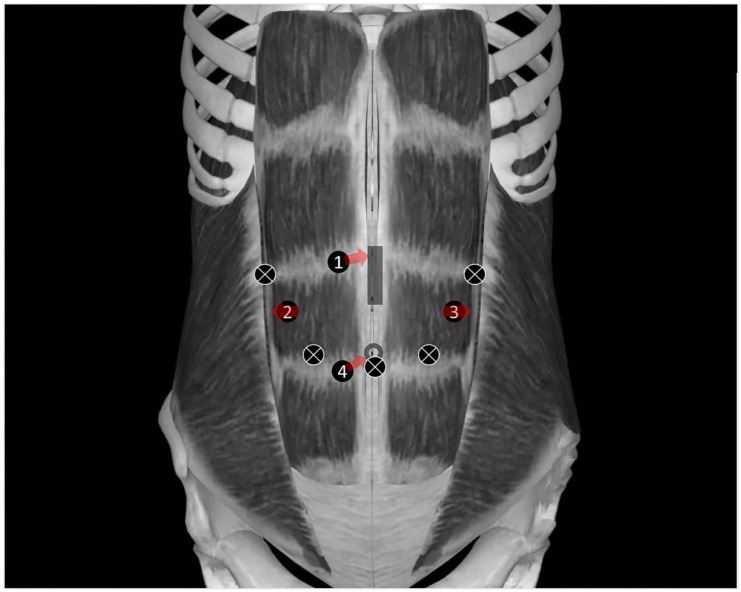
A simple diagram of the location of the median incision, the trocars, and the injection point of the rectus sheath block. Captions: The X represents the insertion point of the trocars, the vertical rectangle represents the median incision, the circle indicates the position of the belly button, numbers 1, 2, 3, and 4 represent the local anesthetic administration point, and the red arrow indicates the direction of the local anesthetic injection.

**Figure 3 jcm-12-01414-f003:**
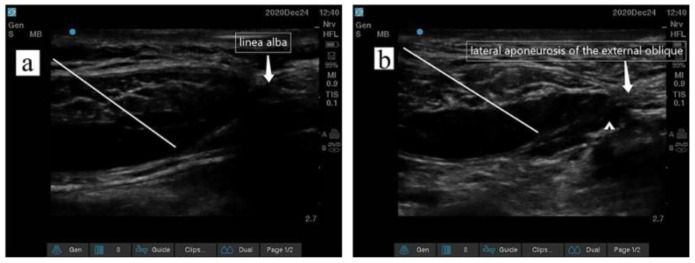
(**a**) represents ultrasound images of the rectus sheath block at puncture points 1 and 4, and (**b**) represents ultrasound images of the rectus sheath block at puncture points 2 and 3. Caption: The white line indicates the position of the puncture needle.

**Table 1 jcm-12-01414-t001:** Demographic characteristics of patients.

	Group IBRSB (n = 30)	Group C (n = 30)	*p* Value
ASA grade (IQR)	2 (2, 3)	2 (2, 3)	0.288
Age (mean, years)	61.87 ± 10.43	59.5 ± 10.78	0.391
BMI (mean, kg/m²)	22.36 ± 4.20	23.75 ± 4.13	0.201
Gender (n)			0.435
Male	19	15	
Female	11	15	
Surgical methods (n)			0.926
LADG	11	12	
LAPG	5	4	
LATG	14	14	
The pathologic types (n)			0.706
Adenocarcinoma	27	25	
Other types of cancer	3	5	

ASA, American Society of Anesthesiologists; BMI, body mass index; Group IBRSB, Group incision-based rectus sheath block (IBRSB) with ropivacaine; Group C, Group IBRSB with normal saline; LADG, laparoscopic-assisted radical distal gastrectomy; LAPG, laparoscopic-assisted proximal gastrectomy; LATG, laparoscopic-assisted total gastrectomy. Values are expressed as mean ± SD, median (interquartile range IQR), and absolute number.

**Table 2 jcm-12-01414-t002:** Surgical methods, intraoperative and immediate postoperative data for the two anesthetic groups.

	Group IBRSB (n = 30)	Group C (n = 30)	*p* Value
Propofol dose (IQR, mg)	850 (757.5, 1040)	895 (830, 1218.5)	0.211
Remifentanil dose (IQR, μg)	504.5 (246.25, 977.75)	2250 (1700, 2575)	<0.001 *
Sufentanil dose (IQR, μg)	32.5 (30, 37.25)	43.75 (40, 45)	<0.001 *
Duration of surgery (mean, min)	230.3 ± 44.2	249.7 ± 55.5	0.141
Duration of anesthesia (IQR, min)	267 (240, 288.75)	277.5 (240, 316.75)	0.313
Eye opeaing time (IQR, min)	19 (14, 20.75)	22.5 (18, 30)	0.007 *
Extubation time (IQR, min)	22 (19, 27.5)	27.5 (20, 35)	0.033 *
Duration of PACU stay (IQR, min)	35 (40, 50)	50 (40, 55.75)	0.031 *
Crystal infusion (IQR, ml)	1600 (1500, 1600)	1600 (1500, 1900)	0.372
Colloid infusion (IQR, ml)	500 (500, 500)	500 (500, 1000)	0.042 *

Group IBRSB, Group incision-based rectus sheath block (IBRSB) with ropivacaine; Group C, Group IBRSB with normal saline; PACU, postanesthesia care unit. Values are expressed as mean ± SD, and median (interquartile range [IQR]) and absolute number. * *p* < 0.05.

**Table 3 jcm-12-01414-t003:** Postoperative data for the two anesthetic groups.

	Group IBRSB (n = 30)	Group C (n = 30)	*p* Value
Postoperative nausea (%)	1 (3.3)	4 (13.3)	0.166
Postoperative vomiting (%)	0 (0)	0 (0)	1.000
Time for the first ambulation after surgery (IQR, h)	79 (69.75, 94.75)	86 (74.25, 92.75)	0.664
Time for the first flatus after surgery (mean, h)	66.53 ± 25.04	77.9 ± 17.37	0.048
Postoperative hospital stay (mean, d)	10.33 ± 2.38	10.27 ± 2.38	0.914
Pain scores at rest (IQR)			
Immediate time after surgery	0 (0, 0)	0 (0, 1)	<0.001 *
6 h after surgery	1 (0, 1)	1 (1, 1)	0.003 *
12 h after surgery	1 (1, 1)	2 (2, 3)	<0.001 *
24 h after surgery	2 (1, 2)	3 (2, 3)	<0.001 *
48 h after surgery	1 (1, 2)	2 (2, 2.75)	0.032 *
Pain scores during activity (IQR)			
Immediate time after surgery	1 (0, 1)	1.5 (1, 2.75)	<0.001 *
6 h after surgery	2 (1.25, 3)	2.5 (2, 3.75)	0.001 *
12 h after surgery	3 (2, 3)	4 (3, 5)	<0.001 *
24 h after surgery	3 (3, 4)	4.5 (4, 5)	<0.001 *
48 h after surgery	3 (3, 3.75)	4 (3.25, 4)	0.016 *
The sufentanil consumption of PCA (IQR, ug/kg)			
24 h after surgery	0.85 (0.75, 0.95)	1.1 (1.05, 1.15)	<0.001 *
48 h after surgery	2.05 (1.95, 2.15)	2.3 (2.25, 2.35)	<0.001 *
Flurbiprofen axetil consumption (IQR, mg)			
The day of surgery	100 (37.5, 100)	100 (50, 100)	0.279
The first day after surgery	100 (87.5, 100)	100 (100, 100)	0.678
The second day after surgery	100 (0, 100)	100 (87.5, 100)	0.313
Patient satisfaction rating (IQR)	4 (4, 5)	2 (2, 3)	<0.001 *

Group IBRSB, Group incision-based rectus sheath block (IBRSB) with ropivacaine; Group C, Group IBRSB with normal saline, PCA, patient-controlled analgesia. Values are expressed as mean ± SD, median (interquartile range [IQR]), and absolute number. * *p* < 0.05.

## Data Availability

The dataset is not publicly available due to privacy and ethical restrictions. The data presented in this study are available on request from the corresponding author.

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
