# Peer review of "Opioid-Sparing Effect of Multi-Point Incision-Based Rectus Sheath Block in Laparoscopic-Assisted Radical Gastrectomy: A Randomized Clinical Trial"

_jcm, 2023, doi:10.3390/jcm12041414_

Round 1
Reviewer 1 Report
Thank you for your referral of this interesting paper on perioperative technique for laparoscopic gastric surgery. This operative procedure is rare in my country but I understand that it is quite common in Asian countries and hence the study is topical.
I have a number of issues - mostly fairly minor - with the research as presented.
The English language used is quite good but in areas the terminology used is a bit odd. Review by a translation service experienced in scientific publishing may be helpful. For example, the term in the introduction, "However, some 50 people thought that..." sounds as if we are talking about our friends in discussion.... More properly one would state "Some researchers report that...." as one assumes they are referring to published research.
Some terms used in the paper need to be defined more clearly in my view. The pain scores, the "recovery" need to be detailed. The exclusion criteria are not worded well and are confusing. In addition:
Pg 2 lines 73 regarding the various "exclusion criteria" - some of these items are not ones which can be identified preoperatively and thus can be 'exclusion' criteria - intraoperative events for example. In particular the comment (line 77) "any condition that the investigator determined would adversely affect the study" cannot be an exclusion criteria - this needs to be specifically identified preoperatively...
Pg4 line 123+ I am confused by the description of the rectus sheath block from the text and the diagram. The text is worded in the singular which suggests that one injection was performed; the diagram suggests 4 injection points, of which two are in the midline and therefore not within the rectus sheath, which does not extend to the midline. Where was the local actually injected? Line 270 does refer to multiple injection points.
Pg 5 line 145 with table 3 - I do not understand the pain score (presumably VAS score in the text) recorded. I am used to scores of 0-10 on a linear analogue score and scores of 0,1,& 2 I have never seen postoperatively, almost as if the patient has not had surgery. This should be clarified.
Table 2 - PACU duration of 35 minutes?? This is very low and I do not undertand how it can be so short after such major surgery. PACU discharge criteria detail would help understand this data, the difference and how the absolute numbers can be achieved. "Recovery time" - ? definition.
Table 3 - I am also impressed by - but don't really understand, and are somewhat suspicious of - other aspects of this data. In particular, by what amounts to the complete absence of nausea and vomiting, particularly as all the patients are receiving narcotics. I've never seen this either. How is this parameter assessed? Also "anal exhaust" - I think we call this flatus - within an hour of the end of the case is unusual - most commonly one talks about days before the first sign of bowel activity such as this.
Page 10 line 290 the authors report that "recovery time" (this term is not defined) and 'extubation time' are different between groups - in table 2 this is recorded numerically and reported to be statistically significant - however the times are actually very similar numerically and I am not sure that there is real clinical significance shown here.
Lines 310 I don't understand this sentence.
Overall the paper is interesting and the success of the local anesthetic technique that the author's used (but defined inadequately) in reducing the narcotic consumption appears valuable so I would be supportative of publishing it if some of the details can be clarified.
Author Response
Dear Editors and Reviewers:
Thank you very much for sending us the valuable comments and critiques of the reviewers on our manuscript “Opioid-sparing effect of multi-point rectus sheath block based on incision location in laparoscopic-assisted radical gastrectomy: a randomized clinical trial”, which is very helpful to improve the quality of our paper. We have revised our manuscript carefully, we changed the title to “Opioid-sparing effect of multi-point incision-based rectus sheath block in laparoscopic-assisted radical gastrectomy: a randomized clinical triall” to express our research clearly and concisely. In order to highlight the difference between the experimental group and the control group, we changed the name of the experimental group “A” to “IBRSB”, and the control group “B” to “C”. We hope that the new version of our paper will meet with your approval.
Following is our point-by-point response to these concerns. We used the “Track Changes” function to make changes to the original for review only. We truly appreciate the hard work and careful consideration of the Editors and Reviewers.
Name: Guang-hong Xu
- mail: [email protected]
Comment: 1)
> The English language used is quite good but in areas the terminology used is a bit odd. Review by a translation service experienced in scientific publishing may be helpful. For example, the term in the introduction, "However, some 50 people thought that..." sounds as if we are talking about our friends in discussion.... More properly one would state "Some researchers report that...." as one assumes they are referring to published research;
Response:
Thank you for your hard work and these constructive suggestions. We have modified the terminology used in areas.
Comment: 2)
> Some terms used in the paper need to be defined more clearly in my view. The pain scores, the "recovery" need to be detailed;
Response:
Thank you for your comment. We added to define these terms more clearly. The visual analogue scale (VAS) pain score was used in our study to evaluate postoperative pain. Shortening the time of PACU stay was defined as beneficial to “recovery”.
Comment: 3)
> The exclusion criteria are not worded well and are confusing. In addition:
Pg 2 lines 73 regarding the various "exclusion criteria" - some of these items are not ones which can be identified preoperatively and thus can be 'exclusion' criteria - intraoperative events for example. In particular the comment (line 77) "any condition that the investigator determined would adversely affect the study" cannot be an exclusion criteria - this needs to be specifically identified preoperatively...;
Response:
Thank you for this important comment. We revised the exclusion criteria. This sentence was revised as follow:
Exclusion criteria were contraindications to RSB, such as coagulopathy, infection at the puncture site, preoperative cognitive impairment, a mental or language barrier; regular opioid usage; the surgical incision or trocar insertion being clearly beyond the range of the rectus sheath; chemotherapy prior to surgery; surgeons were changed; refusal to participate, and any condition that the investigator determined would adversely affect the study.
Comment: 4)
> Pg4 line 123+ I am confused by the description of the rectus sheath block from the text and the diagram. The text is worded in the singular which suggests that one injection was performed; the diagram suggests 4 injection points, of which two are in the midline and therefore not within the rectus sheath, which does not extend to the midline. Where was the local actually injected? Line 270 does refer to multiple injection points.
Response:
Thank you for your suggestion. The description of RSB in the text is not detailed enough. All of the patients received incision-based RSB, there are four injection points in total. Injection points 1 and 4 in Figure 2 are located the position of rectus sheath near midline, the numbers 1 and 4 represent the injection point, and the arrow represents the needle injection direction( We re-uploaded the clearer Figure 2). In the ultrasound image, it is shown that the injection site is in the rectus abdominis sheath, and the needle direction is towards the midline, as shown in Figure 3.
Comment: 5)
> Pg 5 line 145 with table 3 - I do not understand the pain score (presumably VAS score in the text) recorded. I am used to scores of 0-10 on a linear analogue score and scores of 0,1,& 2 I have never seen postoperatively, almost as if the patient has not had surgery. This should be clarified.;
Response:
Thank you for this important comment. We recorded the VAS pain score. Scoring criteria: 0 point represents painless; 1-3 points represent tolerable mild pain; 4-6 points represent pain and affect sleep, but can tolerate; 7-10 points represent severe pain and unbearable. The numbers 0, 1, 2 were the interquartile range of VAS score at rest. We think that the possible reason for the low VAS pain score is that the incision for LAG surgery is smaller and the postoperative pain is less severe. Besides, the multimodal analgesia regimen was adopted to further reduce the postoperative pain of patients, sufentanil, dexmedetomidine and flurbiprofen axetil were used during the operation, and flurbiprofen axetil was used routinely after the operation. PCA was opened with 3mL bolus and, infusion at 2ml/L was continued when the first postoperative VAS score ≥ 3 or if the patient made a request. A rescue analgesic drug of 50–100mg of flurbiprofen axetil was given if pain persisted or if the patients required it ( VAS score > 3 was our writing error. Our study was tested with VAS ≥ 3 and has now been revised to the correct “ VAS score ≥ 3”).
Comment: 6)
> Table 2 - PACU duration of 35 minutes? This is very low and I do not undertand how it can be so short after such major surgery. PACU discharge criteria detail would help understand this data, the difference and how the absolute numbers can be achieved. "Recovery time" - ? definition;
Response:
Thank you for your comment. The possible reasons for the short residence time of PACU are as follows. First, this study adopts a multimodal analgesia program, which minimizes the use of sufentanil and exerts the advantages of ultra-short-acting remifentanil. Second, we used the depth of anesthesia monitoring during the operation, BIS monitoring helps to avoid deep anesthesia, maintain the appropriate depth of anesthesia, and reduce the use of sedative drugs. Third, there were no very elderly and critically ill patients in this study. We supplemented the PACU discharge criteria. When Steward score is greater than or equal to 4, we believe that patients can leave PACU. Steward scoring scale:The degree of wakefulness is 2 points for being completely awake, 1 points for responding to stimulus, and 0 points for not responding to stimulus. The degree of respiratory tract patency was 2 points for coughing and expectorating according to the doctor's orders, 1 point for self-maintenance of respiratory tract patency, and 0 points for the need for support. Body activity, body conscious activity 2 points, body unconscious activity 1 points, body no activity 0 points. Recovery time is the time of eye opening and PACU stay duration. The recovery time in Table 2 is written incorrectly and has been modified to eye opening time.
Comment: 7)
>Table 3 - I am also impressed by - but don't really understand, and are somewhat suspicious of - other aspects of this data. In particular, by what amounts to the complete absence of nausea and vomiting, particularly as all the patients are receiving narcotics. I've never seen this either. How is this parameter assessed? Also "anal exhaust" - I think we call this flatus - within an hour of the end of the case is unusual - most commonly one talks about days before the first sign of bowel activity such as this;
Response:
Thank you for your suggestion. Nausea evaluation criteria : no nausea was 0; nausea, does not affect eating and daily life for 1 point; nausea, affect eating and daily life for 2 points; bedridden with nausea for 3 points. Vomiting evaluation criteria: no vomiting or only mild nausea was 0; vomiting 1-2 times a day was 1 point; vomiting 3-5 times a day was 2 points; vomiting >5 times a day was 3 points. When the score ≥ 1 points for nausea or vomiting. In this study, there was 1 patient with nausea in group IBRSB and 4 patients in group C, no patients with vomiting symptoms were found in either group. The reason may be related to the use of multimodal analgesia in this study, multimodal analgesia can significantly reduce the dosage of opioids, reduces the incidence of nausea and vomiting. It may also be related to the small sample size of this study. And all patients used dexamethasone, which may help reduce the incidence of nausea and vomiting. Postoperative nausea and vomiting were not the main indicators of observation in this study. In Table 3, the units of the time of the first flatus after surgery and the first postoperative getting out of bed activity time are wrong, and the unit has been changed to “h”.
Comment: 8)
> Page 10 line 290 the authors report that "recovery time" (this term is not defined) and 'extubation time' are different between groups - in table 2 this is recorded numerically and reported to be statistically significant - however the times are actually very similar numerically and I am not sure that there is real clinical significance shown here;
Response:
Thank you for your comment. The “Recovery time ” in Table 2 is a written incorrectly. We have changed “Recovery time” to ”Eye opening time”. The data in Table 2 were represented by interquartile range and show that, the time of eye opening, extubation, and PACU stay in group IBRSB were significantly shorter than that in group C (P=0.007, P=0.033, and P=0.031, respectively). In follow-up trials, we will further investigate whether this regimen is clinically meaningful for reducing recovery time.
Comment: 9)
> Lines 310 I don't understand this sentence.
>in our study, the RSB might indeed have a greater advantage compared to patients who did not receive RSB ….
Response:
Thank you for your comment. This sentence was revised as follow:
In our study, IBRSB significantly reduced intraoperative opioid use and improved postoperative analgesia compared with controls, but this study did not investigate whether IBRSB was superior to bilateral RSB. We plan to further investigate the difference between IBRSB and the bilateral RSB in the future.
Hopefully, we have made an appropriate revision based on the comments of the reviewers. We thank you wholeheartedly for your excellent work. Your kind assistance is greatly appreciated. We look forward to any future correspondence.
Best wishes!
Guanghong XU

Reviewer 2 Report
Please find attached the track-changes word file.

Author Response
Dear Editors and Reviewers:
Thank you very much for sending us the valuable comments and critiques of the reviewers on our manuscript “Opioid-sparing effect of multi-point rectus sheath block based on incision location in laparoscopic-assisted radical gastrectomy: a randomized clinical trial”, which is very helpful to improve the quality of our paper. We have revised our manuscript carefully, we changed the title to “Opioid-sparing effect of multi-point incision-based rectus sheath block in laparoscopic-assisted radical gastrectomy: a randomized clinical triall” to express our research clearly and concisely. In order to highlight the difference between the experimental group and the control group, we changed the name of the experimental group “A” to “IBRSB”, and the control group “B” to “C”. We hope that the new version of our paper will meet with your approval.
Following is our point-by-point response to these concerns. We used the “Track Changes” function to make changes to the original for review only. We truly appreciate the hard work and careful consideration of the Editors and Reviewers.
Name: Guang-hong Xu
- mail: [email protected]
Comment: [M1]
> Clarify exclusion.
>In total 84 patients with American Society of Anesthesiologists grades â…¡-â…¢ and who were about to undergo LAG were screened; 63 were aged 18–75 y
Response:
Thank you for your hard work and these constructive suggestions. The exclusion criteria are described in the text. This sentence was revised as follow:
A total of 76 patients were included. Exclusion criteria were contraindications to RSB, such as coagulopathy, infection at the puncture site, preoperative cognitive impairment, a mental or language barrier; regular opioid usage; the surgical incision or trocar insertion being clearly beyond the range of the rectus sheath; chemotherapy prior to surgery; surgeons were changed; refusal to participate, and any condition that the investigator determined would adversely affect the study, a total of 76 patients participated in the study.
Comment: [M2]
> rephrase
>Patients in group A (n=31) received ultrasound-guided RSB, and the patients received 0.4% ropivacaine 40–50mL.
Response:
Thank you for your comment. This sentence was revised as follow:
Patients in group IBRSB (n=38) received ultrasound-guided IBRSB, injected drug totals 0.4% ropivacaine 40-50 mL.
Comment: [M3]
> incorrect ref
>Laparoscopic-assisted radical gastrectomy (LAG) is an effective surgery for the treatment of gastric cancer.
Response:
Thank you for your suggestion. We have revised the references cited here.
1.J. Hallet, S. Labidi, A. Bouchard-Fortier, A. Clairoux, J.P. Gagné. Oncologic specimen from laparoscopic assisted gastrectomy for gastric adenocarcinoma is comparable to D1-open surgery: the experience of a Canadian centre. Can J Surg 2013; 56: 249-55.
Comment: [IL4]
> rephrase – people don’t think – research has been controversial
>some people thought that there was no difference and that the effect was not obvious compared to the control group.
Response:
Thank you for this important comment. This sentence was revised as follow:
Some researches have been controversial that there were no differences and that the effects were not obvious compared to the control group.
Comment: [IL5]
> reference?
>Therefore, only bilateral TAPB or RSB can effectively block incision pain.
Response:
Thank you for this important comment. There is an error in this sentence, this sentence was revised as follow:
Therefore, only bilateral TAPB or RSB may be difficult to effectively block incision pain.[1,2]
- Kinjo, T. Kurita, Y. Fujino, T. Kawasaki, K. Yoshino, T. Hachisuga. Evaluation of laparoscopic-guided rectus sheath block in gynecologic laparoscopy: A prospective, double-blind randomized trial. Int J Surg 2019; 62: 47-53.
- T.K. Oh, S.J. Lee, S.H. Do, I.A. Song. Transversus abdominis plane block using a short-acting local anesthetic for postoperative pain after laparoscopic colorectal surgery: a systematic review and meta-analysis. Surg Endosc 2018; 32: 545-52.
Comment: [IL6]
> Cite Fig.1 here and comment briefly
>63 were aged 18–75 y
Response:
Thank you for your comment. This sentence was revised as follow:
In total, 84 patients with American Society of Anesthesiologists grades â…¡–â…¢, aged 18–75 y, and who were about to undergo LAG were screened. Exclusion criteria were contraindications to RSB, such as coagulopathy, infection at the puncture site, preoperative cognitive impairment, a mental or language barrier; regular opioid usage; the surgical incision or trocar insertion being clearly beyond the range of the rectus sheath; chemotherapy prior to surgery; surgeons were changed; refusal to participate; and any condition that the investigator determined would adversely affect the study, a total of 76 patients participated in the study, as shown in Figure 1.
Comment: [IL7]
>Expected preoperatively or postop exclusion?
>intraoperative blood loss was over 500mL
Response:
Thank you for your comment. We are postoperatively excluded patients with intraoperative blood loss was over 500mL.
Comment: [IL8]
> Check spelling (Chang, Anakysis)
> The CONSORT flowchart
Response:
Thank you for your comment. We have modified the wrong words. "Chang" was amended to "Change" and "Anakysis" was amended to "Analysis".
Comment: [IL9]
>Incorrect ref
>During the surgery, heart rate (HR) or SBP >20% of the baseline value for 1 min and/or BIS>60 were considered as insufficient anesthesia.
Response:
Thank you for your suggestion. We have revised the references cited here.
- Bloc, B.P. Perot, H. Gibert, J.D. Law Koune, Y. Burg, D. Leclerc, et al. Efficacy of parasternal block to decrease intraoperative opioid use in coronary artery bypass surgery via sternotomy: a randomized controlled trial. Reg Anesth Pain Med 2021; 46: 671-8.
- N. Schmidt, P. Bischoff, T. Standl, G. Lankenau, M. Hilbert, J. Schulte Am Esch. Comparative evaluation of Narcotrend, Bispectral Index, and classical electroencephalographic variables during induction, maintenance, and emergence of a propofol/remifentanil anesthesia. Anesth Analg 2004; 98: 1346-53, table of contents.
Comment: [IL10]
> rephrase
> If BIS>60, the bolus dose of propofol was injected with a dose of 0.5mg/kg before increasing the infusion rate of propofol.
Response:
Thank you for your comment. This sentence was revised as follow:
When BIS >60, propofol is given intravenously bolus at a dose of 0.5mg/kg, and then the infusion rate of propofol is adjusted until the BIS range is between 40-60.
Comment: [IL11]
> rephrase
>the bolus dose of remifentanil was firstly injected with a dose of 0.5μg/kg before increasing the infusion rate of remifentanil.
Response:
Thank you for your comment. This sentence was revised as follow:
Refentanil is given as an intravenous bolus at a dose of 0.5μg/kg, followed by an increasing in the rate of remifentanil infusion.
Comment: [IL12]
> adjusted?
>The infusion rate of remifentanil was increased in the range of 0.5μg/kg/h to 1 μg/kg/h and adjust once every 15–20s until the anesthesia was satisfactory.
Response:
Yes,here it is adjusted.
Comment: [IL13]
> What was the goal here?
>….vasoactive drugs were injected….
Response:
Thank you for your comment. When the infusion dose of remifentanil exceeded 1 mg/h, 5–10μg of sufentanil was injected. When the hemodynamic change after sufentanil injection is still higher than 20% of the basal value, the hemodynamic change is considered to be related to the patient's underlying disease such as preoperative hypertension, etc, and vasoactive drugs are given.
Comment: [IL14]
> Was analgesia switched off because the needs were low?
>when the infusion dose of remifentanil was less than 0.25mg/h, the dose was gradually reduced until the infusion was stopped
Response:
Thank you for your comment. Analgesics are switched off because of low demand.This sentence was revised as follow:
When blood pressure stabilizes ± 20% of the basal value, the infusion dose of refentanil is gradually reduced according to changes in blood pressure under the infusion is stopped.
Comment: [IL15]
> For all patients?
>0.7 monitored anesthesia care (MAC) sevoflurane was administered to prevent intraoperative awareness
Response:
Thank you for this important comment. All of the patients received 0.7 minimum alveolar concentration (MAC) sevoflurane was administered to prevent intraoperative awareness.
Comment: [IL16]
>When was this commenced?
>and all of the patients received 50ug of dexmedetomidine at a rate of 0.1–0.3μg/kg/h
Response:
Thank you for the reminder. All of the patients received 50ug of dexmedetomidine at a rate of 0.1–0.3μg/kg/h at the beginning of anesthesia until 30min before the end of surgery.
Comment: [IL17]
> NS?
>(PCA: the drug consisted of 2.5ug/kg of sufentanil and 100mg of flurbiprofen axetil, with a total dose of 100 mL. )
Response:
Thank you for your comment. PCA: the drug consisted of 2.5ug/kg of sufentanil, 100mg of flurbiprofen axetil and NS, with a total dose of 100 mL.
Comment: [IL18]
>X?
>Captions: The cross represents the insertion point of the trocars, the vertical rectangle represents the median incision
Response:
Thank you for your suggestion. We've changed "cross" to "X". The "X" represents the insertion point of the trocars, the vertical rectangle represents the median incision.
Comment: [IL19]
>Number?
>Figure a represents ultrasound images of the rectus sheath block at puncture points 1 and 4, and Figure b represents ultrasound images of the rectus sheath block at puncture points 2 and 3
Response:
Thank you for your constructive comments. We have added the missing number. This sentence was revised as follow:
Figure 3a represents ultrasound images of the rectus sheath block at puncture points 1 and 4, and Figure 3b represents ultrasound images of the rectus sheath block at puncture points 2 and 3.
Comment: [IL20]
> Which Figure?
>Figure a represents ultrasound images of the rectus sheath block at puncture points 1 and 4, and Figure b represents ultrasound images of the rectus sheath block at puncture points 2 and 3
Response:
Thank you for your comment. We have added the missing number. This sentence was revised as follow:
Figure 3a represents ultrasound images of the rectus sheath block at puncture points 1 and 4, and Figure 3b represents ultrasound images of the rectus sheath block at puncture points 2 and 3.
Comment: [IL21]
> What scoring method was used?
>and patient satisfaction.
Response:
Thank you for your comment. We used the Likert 5-level scale (1 being very dissatisfied, 5 being very satisfied). A sentence has been added to the revised manuscript as follow:
Likert 5-level scale , 1 being very dissatisfied, and 5 being very satisfied.
Comment: [IL22]
> Shouldn’t this be in Results section?
>The amounts of remifentanil in group A (RSB) were 444, 470, 911, 392, 450, 850, 2100, 350, 1200, and 1350μg, while the amounts of remifentanil in group B (normal saline control) were 565, 650, 1250, 1800, 1100, 950, 1250, 1850, 2350, and 1850μg. The respective mean values were 851.7 and 1356.5μg. The respective standard deviations were 564.5 and 577.7μg.
Response:
Thank you for your constructive comments. The amount of remifentanil used in the pilot study does not need to be listed, and this section has been removed. Only the mean and standard deviation were retained. The respective mean values were 851.7 and 1356.5μg. The respective standard deviations were 564.5 and 577.7μg.
Comment: [IL23]
> explain
>Our study calls for weak opioid management.
Response:
Thank you for your comment. A sentence has been added to the revised manuscript as follow:
Our study observed whether incision-based RSB can save opioid use in LAG.
Comment: [IL24]
> hypothesize
>We believe that the RSB can produce a partial block effect on the supraperitoneal nerve endings, while the TAPB cannot produce the block effect
Response:
Thank you for the reminder. This sentence was revised as follow:
Based on posterior rectus sheath block closer to the peritoneum, we hypothesize that the RSB could produce a partial block effect on the supraperitoneal nerve endings, while the TAPB may be difficult to produce the block effect.
Hopefully, we have made an appropriate revision based on the comments of the reviewers. We thank you wholeheartedly for your excellent work. Your kind assistance is greatly appreciated. We look forward to any future correspondence.
Best wishes!
Guanghong XU

Reviewer 3 Report
The topic of the article is interesting and actual, but the manuscript needs corrections before publication. First of all, language correction is needed (e.g., line 78, MAC is not monitored anesthesia care, anal exhaust is not a common term, better use dose instead of amounts, etc.).
The text has several methodological flaws. The protocol for postoperative analgesia is not clear. At what intervals was flurbiprofen administered? Common practice is to give NSAIDs in regular doses, and then increase opioids. Table 3 is not clear as well. If the VAS score was 0-1, why were opioids given? The protocol states that they are to be given only when VAS>3. The legend states that the values are written as mean ± standard deviation, but SD is missing.
The discussion states that RSB decreases VAS and increases opioid consumption. How do the authors explain that there is a difference in both parameters? Logically, either we keep the VAS the same and vary the analgesic dose, or we keep the analgesic dose constant and monitor the VAS.
I recommend a major revision before publication.
Author Response
Dear Editors and Reviewers:
Thank you very much for sending us the valuable comments and critiques of the reviewers on our manuscript “Opioid-sparing effect of multi-point rectus sheath block based on incision location in laparoscopic-assisted radical gastrectomy: a randomized clinical trial”, which is very helpful to improve the quality of our paper. We have revised our manuscript carefully, we changed the title to “Opioid-sparing effect of multi-point incision-based rectus sheath block in laparoscopic-assisted radical gastrectomy: a randomized clinical triall” to express our research clearly and concisely. In order to highlight the difference between the experimental group and the control group, we changed the name of the experimental group “A” to “IBRSB”, and the control group “B” to “C”. We hope that the new version of our paper will meet with your approval.
Following is our point-by-point response to these concerns. We used the “Track Changes” function to make changes to the original for review only. We truly appreciate the hard work and careful consideration of the Editors and Reviewers.
Name: Guang-hong Xu
- mail: [email protected]
Comment: 1)
> First of all, language correction is needed (e.g., line 78, MAC is not monitored anesthesia care, anal exhaust is not a common term, better use dose instead of amounts, etc.).
Response:
Thank you for your hard work and these constructive suggestions. We modified the language errors.
Comment: 2)
> The text has several methodological flaws. The protocol for postoperative analgesia is not clear. At what intervals was flurbiprofen administered? Common practice is to give NSAIDs in regular doses, and then increase opioids. Table 3 is not clear as well. If the VAS score was 0-1, why were opioids given? The protocol states that they are to be given only when VAS>3. The legend states that the values are written as mean ± standard deviation, but SD is missing.
Response:
Thank you for this important comment. Flurbiprofen axetil is divided into postoperative conventional analgesia and postoperative rescue analgesic. Flurbiprofen axetil 100 mg was given routinely once a day for the first three days after surgery. If the first postoperative VAS ≥ 3 or the patient had a need, PCA was turned on. If the VAS was still ≥ 3, flurbiprofen axetil 50-100 mg was given as remedial analgesia. VAS score was 0-1, which was the interquartile range at rest. If the first postoperative VAS ≥ 3 in the ward, PCA was turned on ( VAS score > 3 was our writing error. Our study was tested with VAS ≥ 3 and has now been revised to the correct “ VAS score ≥ 3”). "SD" is not missing, these data in figures were originally meant as "mean (SD)", but now they are changed to "mean ± SD".
Comment: 3)
> The discussion states that RSB decreases VAS and increases opioid consumption. How do the authors explain that there is a difference in both parameters? Logically, either we keep the VAS the same and vary the analgesic dose, or we keep the analgesic dose constant and monitor the VAS.
Response:
Thank you for your comment. Our results suggest that incision-based RSB can improve the analgesic effect and reduce the use of opioids, not to increase opioid consumption, and our discussion was based on the results of the trial. Flurbiprofen axetil 100 mg was given routinely once a day for the first three days after surgery. If the first postoperative VAS ≥ 3 or the patient had a need, PCA was turned on. If the VAS was still ≥ 3, flurbiprofen axetil 50-100 mg was given as remedial analgesia.
Hopefully, we have made an appropriate revision based on the comments of the reviewers. We thank you wholeheartedly for your excellent work. Your kind assistance is greatly appreciated. We look forward to any future correspondence.
Best wishes!
Guanghong XU
